# Relationship between occupational dust exposure levels and mental health symptoms among Korean workers

**Wanhyung Lee[1], Jae-Gwang Lee[2], Jin-Ha Yoon[3], June-Hee Lee[2]***

**1** Department of Occupational and Environmental Medicine, Gil Medical Center, Gachon University College of Medicine, Incheon, Republic of Korea, **2** Department of Occupational and Environmental Medicine, Soonchunhyang University Hospital, Seoul, Republic of Korea, **3** Department of Preventive Medicine, Yonsei University College of Medicine, Seoul, Republic of Korea

\* junelee@schmc.ac.kr

## Abstract

Dust and fumes are complex mixtures of airborne gases and fine particles present in all environments inhabited by people. This study investigated the relationship between occupational dust exposure levels and mental health problems such as depression or anxiety, fatigue, and insomnia or sleep disturbance. We analyzed data from the third and fourth Korean Working Conditions Survey (KWCS) conducted by the Korea Occupational Safety and Health Agency in 2011 and 2014. We performed chi-square tests to compare the different baseline and occupational characteristics and mental health status according to occupational dust exposure levels. The odds ratio (OR) and 95% confidence intervals (95% CIs) for mental health symptoms (fatigue, depression or anxiety, and insomnia or sleep disturbance) were calculated using adjusted multiple logistic regression models. A total of 78,512 participants (43,979 in men, 34,533 in women) were included in this study. Among them, 6,013 (7.7%) and 2,625 (3.3%) reported "moderate" and "severe" dust exposure, respectively. Among those who answered "yes" to depression or anxiety, fatigue, insomnia or sleep disturbance, 50 (4.6%), 961 (4.8%), and 123 (5.9%), respectively, demonstrated "severe" occupational dust exposure. Compared to "low" levels of dust exposure, "moderate" and "severe" exposure increased the risk of depression and anxiety (OR = 1.09, 95%CI: 0.88–1.36; OR = 1.16, 95%CI: 0.87–1.58, per exposure respectively); however, this was not statistically significant. For fatigue, significance was observed for "moderate" 1.54 (1.46–1.64) and "severe" 1.65 (1.52–1.80) exposure levels. "Severe" levels increased the risk of insomnia or sleep disturbance (OR = 1.52, 95%CI: 1.25–1.85). These results suggest that the "dust annoyance" concept of mental health, which may be explained by a neurocognitive mechanism, is plausible. Occupational "dust annoyance" has been linked to workers' mental health status, particularly in terms of fatigue and sleep disturbance; a dose-response relationship has been observed. Workers should be protected against dust to support their health and productivity.

**Data Availability Statement:** Data belongs to OSHRI (Korea Occupational Safety & Health Research Institute, oshri.kosha.or.kr) in Korea. All Korean Working Conditions Survey (KCWS) files

are publicly available from the KCWS database. (http://hdl.handle.net/20.500.12236/23243). The authors do not have any special access privileges.

**Funding:** This work was supported by the Soonchunhyang University Research Fund.(JHL) The funder had no role in study design, data collection and analysis, decision to publish, or preparation of the manuscript.

**Competing interests:** The authors have declared that no competing interests exist.

## Introduction

Dust and fumes are complex mixtures of airborne gases and fine particles, which arise from various sources, such as soil, pollution, and are present at working and living environments [1, 2]. Most people are frequently exposed to airborne dust, which they inhale daily, and to dust and fumes transported through the troposphere. Dust and fumes are major contributors to environmental pollution, and their concentration has increased in recent decades as a consequence of rapid industrialization and urbanization [3, 4].

Workers can be protected against inspiring particles released in their occupational environment. Previous research has shown that occupational exposure to dust and fumes can lead to diseases, such as heart or lung disease, or respiratory disorder [5–7], among others. Furthermore, occupational exposure to dusts and fumes was classified as a Group 1 carcinogen by the International Agency for Research on Cancer (IARC) [8, 9]. For example, a previous study has found a significant association between occupational exposure to mineral dust and risk of gastric cancer [10].

Moreover, inflammation caused by dust exposure can have deleterious systemic effects, such as ischemic heart disease, respiratory or digestive system dysfunction [11, 12], and chronic dust exposure can lead to chronic inflammation [13, 14]. In turn, chronic inflammation has been shown to affect the mental health, leading to increase in depression and anxiety, by disrupting hormonal regulation [15–18]. However, few studies to-date have reported on the relationship between dust exposure and mental health. Therefore, this study aimed to investigate the relationship between dust exposure levels and mental health.

## Materials and methods

### Study population

We analyzed data from the third and fourth Korean Working Conditions Survey (KWCS) conducted by the Korea Occupational Safety and Health Agency in 2011 and 2014, respectively. The survey methods and structure used in the KWCS are the same as those used by the European Working Condition Survey (EWCS) for comparing working conditions among countries. The population of the KWCS included a representative sample of current Korean workers aged over 15 years, selected from across the country using multistage systematic cluster sampling. We merged the 2011 and 2014 data sets of the KWCS. However, the KWCS participants differed based on the survey year; the survey structure, which is based on the EWCS, and purpose of the survey, which was to include a representative working population from the Republic of Korea, remained the same in both years. The survey involved face-to-face interviews during house visits, which were conducted by trained interviewers. All participants enrolled in the study agreed to participate in further scientific research. All data are accessible at website: 'http://www.kosha.or.kr/kosha/data/primitiveData.do' A total of 100,039 individuals participated in the third and fourth KWCS (n = 50,032 and n = 50,007, respectively). In the present study, we extracted data on adult participants aged 20–65 years, including information regarding education, household income, symptoms, working duration, and other relevant variables. Finally, data from 78,512 participants (43,979 in men, 34,533 in women) were included in this study after excluding those who were out of the range of age (n = 12,740) and those who missed or refused (n = 8,787) (Fig 1).

### Main variables

The health condition was classified based on self-reported questionnaires, that collected information about symptoms. Depression or anxiety, overall fatigue, and sleep disturbance or

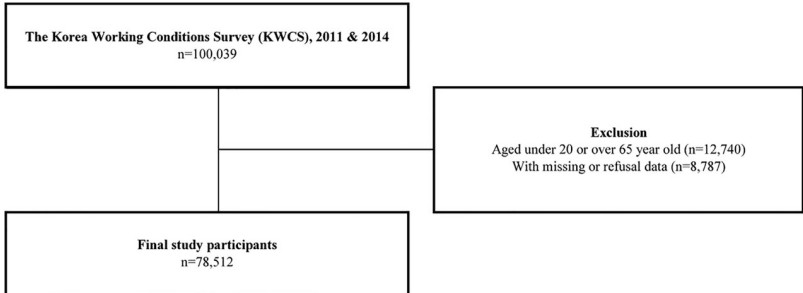

**Fig 1. Schematic diagram depicting study population.**

insomnia were assessed by response to the question: "Did you have any of these health problems over the last 12 months?" This question was identical to the question used in the European Working Conditions Survey. Participants who answered "yes" were considered to have experienced some symptoms.

Additionally, participants were asked the following question regarding occupational dust exposure: "Are you exposed to inhalational smoke, fumes (such as welding or exhaust fumes), powder, or dust (such as wood dust or mineral dust) at work?" Participants answered each question based on a seven-point scale, which represented the following answers: "all of the time," "almost all of the time," "around 3/4 of the time," "around half of the time," "around 1/4 of the time," "almost never," and "never". The responses were divided into three categories: "low" (corresponding to "less than 1/4 of the time"), "moderate" ("around half of the time" or "around 3/4 of the time"), and "severe" (more than 3/4 of the time), with daily working hours used as the reference timeframe for exposure.

## Covariates

Potentially confounding variables included gender, age (<40, 40 to 60, and ≥60), educational level, and household income, and occupational characteristics, such as type of work, size of enterprise, work schedule, and self-rated job satisfaction. The self-rated health status was also included. The educational level was divided into four categories: elementary school and below, middle school, high school, and college or above. Average monthly income was divided into four groups with intervals of 1,000 U.S. dollars. The type of work was also divided into "paid workers" and "others," which included the self-employed, and participants in non-paying occupations, such as homemaking. The size of enterprise was classified based on <1, 2–4, 5–49, and ≥50 workers. Further, we used data on work schedules to identify shift workers, and categorized them into the two following groups based on this information: "shift" and "fixed." Self-rated job satisfaction was divided into three groups based on the answer to the question: "Generally, what do you think about your current job?" The answer "very satisfied" was categorized as "good" job satisfaction, "satisfied" and "not at all satisfied" were into "moderate," and "not very satisfied" was classified as "bad." The self-rated health status was assessed using the question: "How is your health in general?" The responses were grouped as "good" (answers, "very good" and "good"), "moderate" ("fair"), and "bad" ("bad" and "very bad").

## Statistical analysis

We performed chi-square tests to compare the different baseline and occupational characteristics and mental health status based on occupational dust exposure levels. The odds ratio (OR) and 95% confidence intervals (95% CIs) for having mental health problems (fatigue,

depression or anxiety, and insomnia or sleep disturbance) were calculated using adjusted multiple logistic regression models. Potential confounders for the adjusted logistic model were selected by backward stepwise elimination and based on the findings of previous studies [19, 20]. The final multiple logistic model was adjusted for age, sex, education, income, type of work, size of enterprise, work schedule, self-rated health status, and self-rated job satisfaction level.

The weighted prevalence of each mental health problem was estimated based on exposure hours to occupational dust per week, which was converted from daily to weekly exposure hours of occupational dust. All statistical analyses were completed using SAS version 9.4 (SAS Institute Inc., Cary, NC, USA).

## Results

As shown in Table 1, 69,874 (89.0%), 6,013 (7.7%), and 2,625 (3.3%) participants were exposed to "low," "moderate," and "severe" levels of occupational dust. Among the men, 37,632 (85.6%), 4,294 (9.8%), and 2,053 (4.6%) participants were affected by "low," "moderate," and "severe" levels of exposure, respectively; among women, the corresponding proportions were 32,242 (93.4%), 1,719 (5.0%), and 572 (1.6%), respectively. The highest proportion of study participants by socioeconomic characteristics were aged 40~59 years old (50.0%, n = 39,269), were high school graduates (40.5%, n = 31,824), and had monthly incomes of under $1,000 (31.4; n = 24,671). In terms of working characteristics, most participants had fixed working schedules (93.4%, n = 79,325) and moderate job satisfaction levels (92.5%, n = 72,599). The prevalence of depression or anxiety, fatigue, and insomnia or sleep disturbance was 1.4% (n = 1,089), 25.8% (n = 20,276), and 2.6% (n = 2,071), respectively. Those who answered, "yes" to depression or anxiety showed a significant difference across "low" (946; 86.9%), "moderate" (93; 8.5%), and "severe" (50; 4.6%) exposure levels. Those who answered, "yes" to fatigue showed a significant difference across "low" (17,200; 84.8%), "moderate" (2,115; 10.4%), and "severe" (961; 4.8%) levels. Those who answered "yes" to insomnia or sleep disturbance showed a significant difference across "low" (1,783; 86.1%), "moderate" (165; 8.0%), and "severe" (123; 5.9%) exposure levels.

Table 2 demonstrates the results of logistic regression in terms of depression or anxiety for "moderate" (1.09 [0.88–1.36]) and "severe" (1.16 [0.87–1.58]) dust exposure levels when the reference was set to "low". When the "low" exposure group was set as the reference for fatigue, the "moderate" (1.54 [1.46–1.64]) and "severe" (1.65 [1.52–1.80]) groups showed significant results. In terms of insomnia or sleep disturbance, the results were significant in the "moderate" (0.99 [0.84–1.16]) and "severe" (1.52 [1.25–1.85]) groups. The weighted prevalence and linear trends of each mental health symptom according to the exposure hours to occupational dust per week are shown in Fig 2. The linearity in fatigue and insomnia or sleep disturbance indicated the existence of a dose-response relationship.

## Discussion

This study investigated the relationship between occupational dust exposure and workers' mental health, showing an association between symptoms such as fatigue, insomnia or sleep disturbance, and exposure to "moderate" and "severe" levels of dust. These relationships remained when data were adjusted for socio-demographic and occupational characteristics such as age, sex, education, income, and type of work, size of enterprise, work schedule, self-rated health status, and self-rated job satisfaction. Furthermore, the results revealed a dose-response relationship between exposure and outcomes of interest. Therefore, the longer and more pronounced the exposure to dust, the higher was the frequency of mental health

**Table 1. Baseline study participants (n = 78,512) according to occupational dust exposure level.**

| | Total participants n(% of column) | | Occupational dust exposure*, n(% of row) | | | | | |
| | | | Low | | Moderate | | Severe | |
|---|---|---|---|---|---|---|---|---|
| Total subjects | 78.512 | 100.0 | 69,874 | 89.0 | 6,013 | 7.7 | 2,625 | 3.3 |
| Sex | | | | | | | | |
| Men | 43.979 | 56.0 | 37,632 | 85.6 | 4,294 | 9.8 | 2,053 | 4.6 |
| Women | 34.533 | 44.0 | 32,242 | 93.4 | 1,719 | 5.0 | 572 | 1.6 |
| Age | | | | | | | | |
| <40 | 25,432 | 32.4 | 23,448 | 92.2 | 1,369 | 5.4 | 615 | 2.4 |
| <60 | 39,269 | 50.0 | 34,580 | 88.1 | 3,169 | 8.1 | 1,520 | 3.8 |
| ≥60 | 13,811 | 17.6 | 11,846 | 85.8 | 1,475 | 10.7 | 490 | 3.5 |
| Education | | | | | | | | |
| Elementary school | 7,517 | 9.6 | 6,310 | 83.9 | 854 | 11.4 | 353 | 4.7 |
| Middle school | 7,428 | 9.5 | 6,173 | 83.1 | 847 | 11.4 | 408 | 5.5 |
| High school | 31,824 | 40.5 | 27,384 | 86.1 | 3,057 | 9.6 | 1,383 | 4.3 |
| University | 31,743 | 40.4 | 30,007 | 94.5 | 1,255 | 3.9 | 481 | 1.6 |
| Monthly income ($) | | | | | | | | |
| <1,000 | 24,671 | 31.4 | 21,886 | 88.7 | 1,947 | 7.9 | 838 | 3.4 |
| <2,000 | 14,613 | 18.6 | 13,060 | 89.4 | 1,114 | 7.6 | 439 | 3.0 |
| <3,000 | 21,014 | 26.8 | 18,525 | 88.2 | 1,692 | 8.0 | 797 | 3.8 |
| ≥3,000 | 18,214 | 23.2 | 16,403 | 90.1 | 1,260 | 6.9 | 551 | 3.0 |
| Type of work | | | | | | | | |
| Paid workers | 49,289 | 62.8 | 43,764 | 88.8 | 3,624 | 7.4 | 1,901 | 3.8 |
| Self-employed and other | 29,223 | 37.2 | 26,110 | 89.3 | 2,389 | 8.2 | 724 | 2.5 |
| Size of enterprise | | | | | | | | |
| 1 | 19,883 | 25.3 | 17,581 | 88.4 | 1,767 | 9.0 | 535 | 2.6 |
| <5 | 19,462 | 24.8 | 17,715 | 91.0 | 1,287 | 6.6 | 460 | 2.4 |
| <50 | 26,220 | 33.4 | 23,262 | 88.7 | 1,980 | 7.6 | 978 | 3.7 |
| ≥50 | 12,947 | 16.5 | 11,316 | 87.4 | 979 | 9.6 | 652 | 5.0 |
| Work schedule | | | | | | | | |
| Shift | 5,187 | 6.6 | 4,443 | 85.7 | 448 | 8.6 | 296 | 5.7 |
| Fixed | 73,325 | 93.4 | 65,431 | 89.2 | 5,565 | 7.6 | 2,329 | 3.2 |
| Self-rated health status | | | | | | | | |
| Good | 52,826 | 67.3 | 47,555 | 90.0 | 3,651 | 6.9 | 1,620 | 3.1 |
| Moderate | 21,580 | 27.5 | 18,921 | 87.7 | 1,907 | 8.8 | 752 | 3.5 |
| Bad | 4,106 | 5.2 | 3,398 | 82.8 | 455 | 11.0 | 253 | 6.2 |
| Self-rated job satisfaction | | | | | | | | |
| Good | 4,242 | 5.4 | 4,055 | 95.6 | 143 | 3.4 | 44 | 1.0 |
| Moderate | 72,599 | 92.5 | 64,555 | 88.9 | 5,663 | 7.8 | 2,381 | 3.3 |
| Bad | 1,671 | 2.1 | 1,264 | 75.6 | 207 | 12.4 | 200 | 12.0 |
| Depression or anxiety | | | | | | | | |
| Yes | 1,089 | 1.4 | 946 | 86.9 | 93 | 8.5 | 50 | 4.6 |
| No | 77,423 | 98.6 | 68,928 | 89.0 | 5,920 | 7.7 | 2,575 | 3.3 |
| Fatigue | | | | | | | | |
| Yes | 20,276 | 25.8 | 17,200 | 84.8 | 2,115 | 10.4 | 961 | 4.8 |
| No | 58,236 | 74.2 | 52,674 | 90.5 | 3,898 | 6.7 | 1,664 | 2.8 |
| Insomnia or sleep disturbance | | | | | | | | |
| Yes | 2,071 | 2.6 | 1,783 | 86.1 | 165 | 8.0 | 123 | 5.9 |
| No | 76,441 | 97.4 | 68,091 | 89.1 | 5,848 | 7.6 | 2,502 | 3.3 |

*Occupational dust exposure level was categorized by exposure time of daily work hours; low (<50%), moderate (50~75%), and severe (>75%)

**Table 2. Results of odds ratio (OR) and 95% confidence intervals (CI) according to occupational dust exposure level by logistic regression model.**

| | Occupational dust exposure*, OR (95% CI) | | | | | |
| --- | --- | --- | --- | --- | --- | --- |
| | Low | Moderate | | Severe | | P for trend |
| Depression or anxiety | Reference | 1.09 | (0.88–1.36) | 1.16 | (0.87–1.58) | 0.2062 |
| Fatigue | Reference | 1.54 | (1.46–1.64) | 1.65 | (1.52–1.80) | < .0001 |
| Insomnia or sleep disturbance | Reference | 0.99 | (0.84–1.16) | 1.52 | (1.25–1.85) | 0.0010 |

*Occupational dust exposure level was categorized by exposure time of daily work hours; low (<50%), moderate (50~75%), and severe (>75%)

All models are adjusted age, sex, education, income, type of work, size of enterprise, work schedule, self-rated health status, and self-rated job satisfaction level.

symptoms such as fatigue and insomnia, among surveyed workers. However, the incidence of depression or anxiety was not affected by dust exposure. This finding is inconsistent with that of a previous study on past occupational dust exposure among retired Chinese factory workers, where depressive symptoms and anxiety were associated with dust exposure [21].

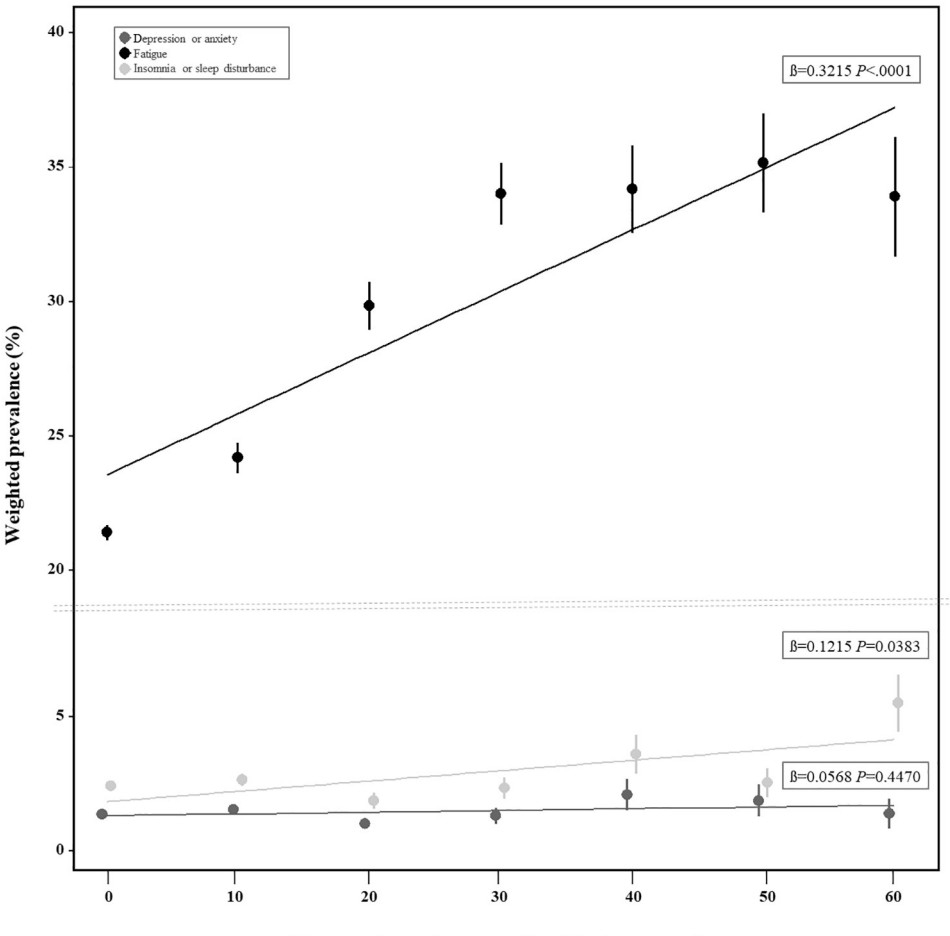

**Fig 2. Weighed prevalence and trend of depression or anxiety, fatigue, and insomnia or sleep disturbance in the Korean workers according to exposure hours to occupational dust per a week.** (All working hours are estimated 60 hours per a week).

An inflammatory response is considered to be the basis for the pathogenesis of various diseases. A significant number of previous studies have focused on elucidating the link between dust exposure and airway inflammation [22–25]. While dust inhalation can cause local inflammation of the airways, other research has shown that it can also lead to systemic inflammation, whereby the inhaled dust enters blood circulation, resulting in oxidative stress and subsequent systemic inflammation [26, 27]. Elevated inflammatory markers have also been found to be closely associated with psychiatric disorders such as major depression [28], anxiety [29], and bipolar disorder [30]. In addition, further studies have shown a relationship between inflammation and symptoms consistent with mental illness, including sleep disturbance and fatigue [31, 32]. A current hypothesis proposed as an explanation for the influence of dust exposure on mental health suggests that inflammation (local or systemic) caused by inhaled dust may be prolonged and may become chronic, resulting in psychiatric problems.

Similar to the mechanism whereby noise annoyance may lead to psychological problems by provoking sustained central autonomic arousal and disruption of the dopamine pathway [33], occupational dust exposure may lead to mental health problems by triggering sustained nervous stimulation, which is associated with cortical activation. The impact of dust exposure on mental health may therefore be referred to as "dust annoyance."

The concept of "dust annoyance" in mental health may be explained by a neurocognitive mechanism. The predisposing factors, such as stressors at the workplace, which are associated with somatic, cognitive, or cortical activation, are closely linked to perpetuating factors such as extension of time in bed due to obstacles to de-arousal from cortical arousal [34]. The concept proposes that "dust annoyance" could continually increase cortical arousal, leading to mental health deterioration, which ranges from symptoms of fatigue and sleep disturbance to depression or anxiety; the current results are in agreement with this concept.

The present study has several limitations. First, owing to the cross-sectional study design, the results indicate an association between "dust annoyance" and the mental health of workers; no conclusions may therefore be drawn with respect to causality. Nevertheless, a biological gradient, which constitutes one of Hill's criteria for causality [35], was detected in our analysis, which indicates that the reported association was causal. However, any such association should be interpreted with caution as most dose-response curves are non-linear owing to complex factors that affect the shape of the curve [36]. Second, as our study was based on self-reported symptoms from questionnaire data, which relies on the accuracy of participants' memory, there was a possibility of recall bias. In addition, the mental health status surveyed in the present study, such as depression or anxiety, fatigue, and insomnia or sleep disturbance was based on self-reported information; therefore, any suspected or reported psychopathologies did not necessarily meet the diagnostic criteria for particular medical conditions.

Third, the type and quantity of inhaled dust was not assessed quantitatively owing to a lack of relevant data. As these factors may have distinct health effects, prospective studies are needed to elucidate the dose-response phenomenon more clearly. Furthermore, we evaluated the mental health status with particular focus on occupational dust exposure levels without considering other health behaviors such as smoking, alcohol drinking, or exercise. Mental health is known to be closely related with health behavior [37]. Unfortunately, health behavioral factors linked to mental health were not accessible owing to the nature of the KWCS data. Lastly, we used occupational dust exposed levels during the working period as the main risk factor for mental health. However, we could not evaluate the conditions during the entire working period, that may have had an adverse impact on the mental health of workers. In this cohort, workers' mental health was closely related with exposure to multiple occupational risk factors at the workplace [38]. Unfortunately, this study does not reflect the conditions of the entire working period. Further studies are needed to investigate this important issue.

## Conclusion

In conclusion, occupational "dust annoyance" in this cohort was linked to workers' mental health status, particularly with fatigue and sleep disturbance, indicating a dose-response relationship. Since conditions such as fatigue and sleep disturbance may decrease work efficiency and lead to injuries at workplaces, it is essential to limit dust exposure in work environments, and to provide personal protective gear to workers.

## Author Contributions

**Conceptualization:** Wanhyung Lee, Jin-Ha Yoon, June-Hee Lee.

**Data curation:** Wanhyung Lee, Jin-Ha Yoon, June-Hee Lee.

**Formal analysis:** Wanhyung Lee.

**Investigation:** Jae-Gwang Lee, Jin-Ha Yoon.

**Methodology:** Wanhyung Lee, Jin-Ha Yoon, June-Hee Lee.

**Resources:** Jae-Gwang Lee.

**Supervision:** Jin-Ha Yoon, June-Hee Lee.

**Validation:** Jae-Gwang Lee, Jin-Ha Yoon, June-Hee Lee.

**Writing – original draft:** Wanhyung Lee, Jae-Gwang Lee, June-Hee Lee.

**Writing – review & editing:** Jae-Gwang Lee, Jin-Ha Yoon, June-Hee Lee.

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
