## [Decision Letter · Decision Letter 0]

27 Nov 2019

PONE-D-19-23328

The effect of occupational dust exposure on Korean workers’ mental health

PLOS ONE

Dear M.D.,Ph.D Lee,

Thank you for submitting your manuscript to PLOS ONE. After careful consideration, we feel that it has merit but does not fully meet PLOS ONE’s publication criteria as it currently stands. Therefore, we invite you to submit a revised version of the manuscript that addresses the points raised during the review process.

We would appreciate receiving your revised manuscript by Dec 30 2019 11:59PM. To enhance the reproducibility of your results, we recommend that if applicable you deposit your laboratory protocols in protocols.io, where a protocol can be assigned its own identifier (DOI) such that it can be cited independently in the future. For instructions see: http://journals.plos.org/plosone/s/submission-guidelines#loc-laboratory-protocols

We look forward to receiving your revised manuscript.

Kind regards,

Michio Murakami

Academic Editor

PLOS ONE

Journal Requirements:

1. We suggest you thoroughly copyedit your manuscript for language usage, spelling, and grammar. If you do not know anyone who can help you do this, you may wish to consider employing a professional scientific editing service.  

2. Please include your tables as part of your main manuscript and remove the individual files. Please note that supplementary tables (should remain/ be uploaded) as separate "supporting information" files

3. We note you have included a table to which you do not refer in the text of your manuscript. Please ensure that you refer to Tables 1&2 in your text; if accepted, production will need this reference to link the reader to the Table.

4. Please ensure that you refer to Figure 2 in your text as, if accepted, production will need this reference to link the reader to the figure.

Additional Editor Comments (if provided):

1. As described in the method, occupational dust exposure is based on the working time. This study showed that working time in dust exposure condition affected rather than the exposure itself affected the mental health. The words “occupational dust exposure” is misleading. Please change these words carefully throughout the manuscript including title.

2. While several variables related to the working condition are included as covariates, the authors did not include the non-exposure working time (i.e., the working time without dust exposure) as covariates. There is a possibility that the other working time rather than the working time in dust exposure condition gave adverse effects on mental health. The authors should discuss this point.

3. The authors did not mention Tables and Figures in the text of the manuscript. Please refer them in relevant positions.

4. Abstract (P2L36-41): The results were not significant. The word “increased” is misleading.

5. Reference lists appeared twice. Please correct them.

Reviewers' comments:

Reviewer's Responses to Questions

**Comments to the Author**

1. Is the manuscript technically sound, and do the data support the conclusions?

Reviewer #1: Yes

Reviewer #2: Yes

Reviewer #3: Partly

2. Has the statistical analysis been performed appropriately and rigorously? 

Reviewer #1: Yes

Reviewer #2: Yes

Reviewer #3: No

3. Have the authors made all data underlying the findings in their manuscript fully available?

Reviewer #1: Yes

Reviewer #2: Yes

Reviewer #3: No

4. Is the manuscript presented in an intelligible fashion and written in standard English?

Reviewer #1: Yes

Reviewer #2: Yes

Reviewer #3: Yes

5. Review Comments to the Author

Reviewer #1: The paper investigates associations between dust exposure and mental health. The study population is a large population-based sample that seems to be representative of the current Korean workforce. Mental Health and dust exposure are both self-reported, but the limitations of this is nicely discussed, as are the results in general. However, since the study sample are of current workers, it would be nice with a paragraph discussing the healthy worker effect and if this could have any effect on the results.

Other minor comments:

I can't find any references to figure 2 in the text

Page 5, lines 78-79: the sentence "the quality of the KWCS" is incomplete.

There are some references in separate brackets where they should be in one bracket, such as page 4 lines 62, 63 and 65.

Reviewer #2: Well done! I think this is a good research study that addresses an underemphasized connection between occupational and mental health.

There are some sections of the discussion that I think could be better phrased to aid with clarity. Line 182-184 is not entirely clear in meaning" "The predisposing factors, such as stressors at a workplace, which are associated with somatic, cognitive, or cortical activation closely linked to perpetuating factors such as extension of time in bed due to obstacle to de-arousal from the cortical arousal." Please re-phrase this.

Excellent job with the limitations of the study.

Reviewer #3: SUMMARY

The authors examined the relationship between subjective occupational dust exposure and mental health conditions (such as depression and anxiety, fatigue, and sleep problems) using representative large-scale survey (KWCS) conducted in 2011 and 2014 with cross-sectional design. As a result of multiple logistic regression, occupational dust exposure was significantly related to fatigue and sleep problems after adjusting key covariates. The paper is generally well written. I recommend some revisions to enhance the clarify of the paper regarding following points:

1. L71-83 Study population:

- Please add availability of the data.

- Please specify the information regarding missing observation.

- Please describe the rational of the integration of data in 2011 and 2014.

2, L86-100 main variables:

- Please describe clearly how depression, anxiety, fatigue, and sleep problems were specified.

- Mental health status was assessed by single item binary items. Please add the validity of the items in terms of ability for screening mental health.

- You mentioned that “most people are unaware that they daily inhale dust” in introduction section. However, occupational dust exposure was assessed using subjective rating. Please describe the validity of subjective rating item of occupational dust exposure used in this study.

3. L102-119. Covariates:

- Please clarify the rationale of including covariates into regression model as potential confounder. Please cite previous studies to explain whether each covariate is valid as a potential confounder.

- Smoking has been shown to be a predictor of onset of anxiety and mood disorders (see, Mojtabai, R., & Crum, R. M. (2013). Cigarette smoking and onset of mood and anxiety disorders. American journal of public health, 103(9), 1656-1665.). Therefore, smoking status can be an important potential confounder for the relationship between occupational dust exposure and mental health. However, smoking status was not included in this study. This point should be mentioned in the discussion section as a critical limitation of this study.

4. Table 1 and L132-141, results:

- I recommend that you do not perform statistical tests on the crosstabulation table in Table 1, in terms of multiplicity of tests.

- It would be more meaningful to show how the proportions and values differ between groups, rather than only mention statistical significance in the result section manuscript related to Table 1.

5. fig.2 and analysis method:

- In figure 2, the relationship between fatigue and exposure hours to occupational dust per week seems to be nonlinear (linearly increasing from 0 to 30 and constant after 30). The analytical method should be reconsidered to clarify this nonlinear relationship.

6. PLOS authors have the option to publish the peer review history of their article (what does this mean?). If published, this will include your full peer review and any attached files.

Reviewer #1: No

Reviewer #2: No

Reviewer #3: Yes: Yoshitake Takebayashi

---

## [Author Response · Author response to Decision Letter 0]

3 Jan 2020

Editor Comments:

1. As described in the method, occupational dust exposure is based on the working time. This study showed that working time in dust exposure condition affected rather than the exposure itself affected the mental health. The words “occupational dust exposure” is misleading. Please change these words carefully throughout the manuscript including title.

Answer) As suggested, we revised the text to indicate the relationship with the levels of exposure instead of the exposure itself. The title has accordingly been revised to: “Relationship between occupational dust exposure levels and mental health symptoms among Korean workers.” 

2. While several variables related to the working condition are included as covariates, the authors did not include the non-exposure working time (i.e., the working time without dust exposure) as covariates. There is a possibility that the other working time rather than the working time in dust exposure condition gave adverse effects on mental health. The authors should discuss this point.

Answer) The current investigation focused on the symptoms of mental health related to occupational dust exposure levels. However, we agree that the conditions of workplace for the entire time spent working has an impact on workers’ health. In this study, we found a significant association between occupational dust exposure levels and symptoms of mental health after adjusting for potential confounders as far as practicable; however, it is important to understand the clinical implications of the conditions during the entire working period on the health status of workers. We have therefore revised the limitations section as follows:

Lastly, we used occupational dust exposed levels during the working period as the main risk factor for mental health. However, we could not evaluate the conditions during the entire working period, that may have had an adverse impact on the mental health of workers. In this cohort, workers’ mental health was closely related with exposure to multiple occupational risk factors at the workplace [1]. Unfortunately, this study does not reflect the conditions of the entire working period. Further studies are needed to investigate this important issue.

3. The authors did not mention Tables and Figures in the text of the manuscript. Please refer them in relevant positions.

Answer) We apologize for the error, and have mentioned the relevant Table and Figure numbers in the text of the Results section as follows: 

As shown in Table 1, 69,874 (89.0%), 6,013 (7.7%), and 2,625 (3.3%) participants were exposed to “low,” “moderate,” and “severe” levels of occupational dust.

Table 2 demonstrates the results of logistic regression in terms of depression or anxiety for “moderate” (1.09 [0.88-1.36]) and “severe” (1.16 [0.87-1.58]) dust exposure levels when the reference was set to “low”.

The weighted prevalence and linear trends of each mental health symptom according to the exposure hours to occupational dust per week are shown in Figure 2. The linearity in fatigue and insomnia or sleep disturbance indicated the existence of a dose-response relationship.

4. Abstract (P2L36-41): The results were not significant. The word “increased” is misleading.

Answer) This study found that except for moderate exposure in insomnia or sleep disturbance (OR 0.99), moderate and severe occupational dust exposure levels did increase the risk of prevalence of each mental health symptom compared to low levels; however, statistical significance was observed for fatigue with moderate and severe exposure and for insomnia or sleep disturbance with moderate exposure levels. We have revised the study results section in the Abstract to improve clarity as follows:

Compared to “low” levels of dust exposure, “moderate” and “severe” exposure increased the risk of depression and anxiety (OR=1.09, 95%CI: 0.88-1.36; OR=1.16, 95%CI: 0.87-1.58, per exposure respectively); however, this was not statistically significant.

5. Reference lists appeared twice. Please correct them.

Answer) We apologize for the error, and have deleted the duplicated references.

 

Review Comments to the Author

Reviewer #1: The paper investigates associations between dust exposure and mental health. The study population is a large population-based sample that seems to be representative of the current Korean workforce. Mental Health and dust exposure are both self-reported, but the limitations of this is nicely discussed, as are the results in general. However, since the study sample are of current workers, it would be nice with a paragraph discussing the healthy worker effect and if this could have any effect on the results.

Answer) Both, the EWCS and KWCS were established to understand the working condition of workers from the ecological perspective. As correctly observed, the healthy worker effect could be an extremely important potential confounder while comparing the working population with the general population or when studying serious conditions such as cancer or death among workers. However, the current investigation exclusively focused on the self-rated mental health status among workers based on dust exposure. 

Other minor comments:

I can't find any references to figure 2 in the text

Page 5, lines 78-79: the sentence "the quality of the KWCS" is incomplete.

There are some references in separate brackets where they should be in one bracket, such as page 4 lines 62, 63 and 65.

Answer) As suggested, we have added the reference to Figure 2 in the Results section, and have corrected the sentence in lines 78-79 on page 5. In addition, all adjacent references cited in separate brackets were merged.

 

Reviewer #2: Well done! I think this is a good research study that addresses an underemphasized connection between occupational and mental health.

There are some sections of the discussion that I think could be better phrased to aid with clarity. Line 182-184 is not entirely clear in meaning" "The predisposing factors, such as stressors at a workplace, which are associated with somatic, cognitive, or cortical activation closely linked to perpetuating factors such as extension of time in bed due to obstacle to de-arousal from the cortical arousal." Please re-phrase this.

Answer) As suggested, we have rephrased the sentence in lines 182-184 to improve clarity as follows:

Similar to the mechanism whereby noise annoyance may lead to psychological problems by provoking sustained central autonomic arousal and disruption of the dopamine pathway [33], occupational dust exposure may lead to mental health problems by triggering sustained nervous stimulation, which is associated with cortical activation.

Excellent job with the limitations of the study.

Answer) Thank you for your encouraging comment. 

 

Reviewer #3: SUMMARY

The authors examined the relationship between subjective occupational dust exposure and mental health conditions (such as depression and anxiety, fatigue, and sleep problems) using representative large-scale survey (KWCS) conducted in 2011 and 2014 with cross-sectional design. As a result of multiple logistic regression, occupational dust exposure was significantly related to fatigue and sleep problems after adjusting key covariates. The paper is generally well written. I recommend some revisions to enhance the clarify of the paper regarding following points:

1. L71-83 Study population:

- Please add availability of the data.

- Please specify the information regarding missing observation.

- Please describe the rational of the integration of data in 2011 and 2014.

Answer) We have revised the methods section with available data from the KWCS site. The data of the study participants in Figure 1 have been revised. The data structure of the KWCS and the integration methods were also revised. 

2, L86-100 main variables:

- Please describe clearly how depression, anxiety, fatigue, and sleep problems were specified.

- Mental health status was assessed by single item binary items. Please add the validity of the items in terms of ability for screening mental health.

Answer) Both, the EWCS and KWCS were established to understand the working conditions of workers from an ecological perspective. The health problem questionnaires were not based on clinical indices, but on self-answered symptoms. Although these do not adequately assess the symptoms of mental health in the clinic, they are useful for evaluating the working condition-related mental health status. We have revised the title to add the word ‘symptoms,’ and have added an explanation in the limitations section as follows:

Second, as our study was based on self-reported symptoms from questionnaire data, which relies on the accuracy of participants’ memory, there was a possibility of recall bias. In addition, the mental health status surveyed in the present study, such as depression or anxiety, fatigue, and insomnia or sleep disturbance was based on self-reported information; therefore, any suspected or reported psychopathologies did not necessarily meet the diagnostic criteria for particular medical conditions.

- You mentioned that “most people are unaware that they daily inhale dust” in introduction section. However, occupational dust exposure was assessed using subjective rating. Please describe the validity of subjective rating item of occupational dust exposure used in this study.

Answer) We have revised the sentence. 

3. L102-119. Covariates:

- Please clarify the rationale of including covariates into regression model as potential confounder. Please cite previous studies to explain whether each covariate is valid as a potential confounder.

Answer) As suggested, we have explained the rationale of including covariates as potential confounders in the methods section, as follows:

Potential confounders for the adjusted logistic model were selected by backward stepwise elimination and based on the findings of previous studies [19, 20]. The final multiple logistic model was adjusted for age, sex, education, income, type of work, size of enterprise, work schedule, self-rated health status, and self-rated job satisfaction level.

- Smoking has been shown to be a predictor of onset of anxiety and mood disorders (see, Mojtabai, R., & Crum, R. M. (2013). Cigarette smoking and onset of mood and anxiety disorders. American journal of public health, 103(9), 1656-1665.). Therefore, smoking status can be an important potential confounder for the relationship between occupational dust exposure and mental health. However, smoking status was not included in this study. This point should be mentioned in the discussion section as a critical limitation of this study.

Answer) Smoking status is a very important risk factor for mental health status. However, the KWCS did not have any information regarding individual health related behaviors such as smoking, drinking, or exercise levels. We could not control all the potential confounders owing to the nature of the data. We asked the OSHA, which is conducting the KWCS, to develop a survey structure focused on health related factors. As suggested, we mentioned this limitation in the manuscript as follows:

Furthermore, we evaluated the mental health status with particular focus on occupational dust exposure levels without considering other health behaviors such as smoking, alcohol drinking, or exercise. Mental health is known to be closely related with health behavior [37]. Unfortunately, health behavioral factors linked to mental health were not accessible owing to the nature of the KWCS data.

4. Table 1 and L132-141, results:

- I recommend that you do not perform statistical tests on the crosstabulation table in Table 1, in terms of multiplicity of tests.

- It would be more meaningful to show how the proportions and values differ between groups, rather than only mention statistical significance in the result section manuscript related to Table 1.

Answer) We have revised table 1 and the results section, as suggested.

5. fig.2 and analysis method:

- In figure 2, the relationship between fatigue and exposure hours to occupational dust per week seems to be nonlinear (linearly increasing from 0 to 30 and constant after 30). The analytical method should be reconsidered to clarify this nonlinear relationship.

Answer) As suggested, we conducted trend analysis of the mental health status according to weekly exposure (in hours) to occupational dust. The relationship between mental health status and occupational dust exposure levels was found to be variable. We therefore calculated the trend of weighted prevalence for the mental health status based on the occupational dust exposure level linearity, using the estimate of beta, statistical significance, and visual graphics. Since ‘linear trend’ was a misleading expression, we have edited the figure legend and related text, accordingly. 

 

1. Lee, S., et al., Symptoms of nervous system related disorders among workers exposed to occupational noise and vibration in Korea. Journal of occupational and environmental medicine, 2017. 59(2): p. 191-197.

2. Park, S., J.-H. Lee, and W. Lee, The Effects of Workplace Rest Breaks on Health Problems Related to Long Working Hours and Shift Work among Male Apartment Janitors in Korea. Safety and Health at Work, 2019.

3. Kang, D., et al., Anxiety, Depression and Sleep Disturbance among Customer-Facing Workers. Journal of Korean Medical Science, 2019. 34(48).

4. Mojtabai, R. and R.M. Crum, Cigarette smoking and onset of mood and anxiety disorders. American journal of public health, 2013. 103(9): p. 1656-1665.

---

## [Decision Letter · Decision Letter 1]

27 Jan 2020

Relationship between occupational dust exposure levels and mental health symptoms among Korean workers

PONE-D-19-23328R1

Dear Dr. Lee,

We are pleased to inform you that your manuscript has been judged scientifically suitable for publication and will be formally accepted for publication once it complies with all outstanding technical requirements.

With kind regards,

Michio Murakami

Academic Editor

PLOS ONE

Additional Editor Comments (optional):

Reviewers' comments:

Reviewer's Responses to Questions

**Comments to the Author**

1. If the authors have adequately addressed your comments raised in a previous round of review and you feel that this manuscript is now acceptable for publication, you may indicate that here to bypass the “Comments to the Author” section, enter your conflict of interest statement in the “Confidential to Editor” section, and submit your "Accept" recommendation.

Reviewer #3: All comments have been addressed

2. Is the manuscript technically sound, and do the data support the conclusions?

Reviewer #3: Yes

3. Has the statistical analysis been performed appropriately and rigorously? 

Reviewer #3: Yes

4. Have the authors made all data underlying the findings in their manuscript fully available?

Reviewer #3: Yes

5. Is the manuscript presented in an intelligible fashion and written in standard English?

Reviewer #3: Yes

6. Review Comments to the Author

Reviewer #3: It seems the revised manuscript reflect all reviewer's comment appropriately and it meet the standard for scientific publication.

7. PLOS authors have the option to publish the peer review history of their article (what does this mean?). If published, this will include your full peer review and any attached files.

Reviewer #3: Yes: Yoshitake Takebayashi

---

## [Editor Report · Acceptance letter]

7 Feb 2020

PONE-D-19-23328R1 

Relationship between occupational dust exposure levels and mental health symptoms among Korean workers 

Dear Dr. Lee:

I am pleased to inform you that your manuscript has been deemed suitable for publication in PLOS ONE. Congratulations! Your manuscript is now with our production department. 

With kind regards,

on behalf of

Dr. Michio Murakami 

Academic Editor

PLOS ONE